# Cooperative Task Execution for Object Detection in Edge Computing: An Internet of Things Application

Petros Amanatidis [1], Dimitris Karampatzakis [1,*], George Iosifidis [2], Thomas Lagkas [1] and Alexandros Nikitas [3,*]

1. Department of Computer Science, International Hellenic University, Agios Loukas, 65404 Kavala, Greece; peamana@cs.ihu.gr (P.A.); tlagkas@cs.ihu.gr (T.L.)
2. Delft University of Technology, Van Mourik Broekmanweg 6, 2628 XE Delft, The Netherlands; g.iosifidis@tudelft.nl
3. Department of Logistics, Marketing, Hospitality and Analytics, Huddersfield Business School, University of Huddersfield, Huddersfield HD1 3DH, UK
* Correspondence: dkara@cs.ihu.gr (D.K.); a.nikitas@hud.ac.uk (A.N.)

**Abstract:** The development of computer hardware and communications has brought with it many exciting applications in the Internet of Things. More and more Single Board Computers (SBC) with high performance and low power consumption are used to infer deep learning models at the edge of the network. In this article, we investigate a cooperative task execution system in an edge computing architecture. In our topology, the edge server offloads different workloads to end devices, which collaboratively execute object detection on the transmitted sets of images. Our proposed system attempts to provide optimization in terms of execution accuracy and execution time for inferencing deep learning models. Furthermore, we focus on implementing new policies to optimize the E2E execution time and the execution accuracy of the system by highlighting the key role of effective image compression and the batch sizes (splitting decisions) received by the end devices from a server at the network edge. In our testbed, we used the You Only Look Once (YOLO) version 5, which is one of the most popular object detectors. In our heterogeneous testbed, an edge server and three different end devices were used with different characteristics like CPU/TPU, different sizes of RAM, and different neural network input sizes to identify sharp trade-offs. Firstly, we implemented the YOLOv5 on our end devices to evaluate the performance of the model using metrics like Precision, Recall, and mAP on the COCO dataset. Finally, we explore optimal trade-offs for different task-splitting strategies and compression decisions to optimize total performance. We demonstrate that offloading workloads on multiple end devices based on different splitting decisions and compression values improves the system's performance to respond in real-time conditions without needing a server or cloud resources.

**Keywords:** edge AI; task offloading; YOLOv5; edge computing; object detection system; Internet of Things

## 1. Introduction

The swift advancement of computing resources and storage of the Internet of Things devices has made significant progress in a wide range of applications, leading to an era in which computing and processing of data are flowing from the cloud to the edge [1,2]. Many cloud-based applications that were executed previously in the cloud are now delivered by edge devices such as tablets, wearable IoT devices, or smartphones [3]. According to recent literature [4,5] the number of connected devices is expected to reach 150 billion by the year 2025. Moreover, almost 70% of the computing workloads will be executed at the edge of the network in the next years. The concept of edge computing involves performing data processing at the edge devices, resulting in improving the performance of the applications by using an edge server which provides many benefits like reducing computational latency

and energy consumption. A great effort has been made by many researchers to develop different optimization tools to minimize energy consumption and latency and to maximize the performance of edge computing applications.

In addition, Artificial Intelligence, defined as a powerful vehicle for the optimization of machines' interpretation, learning, and task-performing capacity [6], has already expanded in a wide range of applications, including reinforcement learning and computer vision. Furthermore, real-time applications like autopilot-assisted driving and, in a few years, connected and autonomous vehicles depend on the fast processing of incoming data, which makes Cloud AI not the best solution. Edge AI is a promising solution to face these problems [7,8]. This solution accommodates not only the inference but also the training of the AI models to the edge. Avoiding the transmission of the data to the cloud mitigates latency, privacy, and traffic load problems. The training of a deep learning model requires intensive computing resources. Nonetheless, the explosive evolution of the hardware and the processing power of the edge devices can provide these intensive computing resources which are required to develop an AI application at the edge [9,10].

The core concept behind edge computing is to provide Cloud-like functionalities closer to the end device and the user. In other words, a Cloud-like device should be capable of transferring (offloading) the computationally-intensive task to a close-edge device, to reduce, for example, latency and power consumption. This approach is also called task offloading, which allows lower latency, improved energy consumption, and much better reliability. Task offloading has gained much attention from the academic community and the industry. Many studies were published in recent years addressing task offloading, for example, Wang et al. in 2017 and Ai Yuan et al. in 2018 made an effort to classify the various types of task offloading. The aspects on which this body of literature is particularly focused are architecture, resource allocation, communications, algorithms, etc. [11–14].

In this paper, we study a cooperative task execution of an edge-assisted object detection system. In more detail, we chose to use the official pre-trained YOLOv5 [15] object detector using the COCO dataset as workload. This was offloaded to every end device to get the object detector evaluation metrics like mAP, precision, recall, and also execution time, and other metrics for every end device. We used the YOLO object detector and the COCO dataset because they can bench-mark the performance of real-time object detection systems. Moreover, we study two natural criteria for selecting how to compress the images and how to split the image set across the available end devices. We searched experimentally for the optimal trade-off among these metrics to provide optimization in terms of execution accuracy and execution time. In our testbed, we highlight optimal trade-offs between the E2E execution time and execution accuracy of our object detection system. We also show the effect of image compression on transmission delay and, of course, execution time. Our main contribution after taking our experimental results and measurements is to develop a statistical model based on our performance metrics which can be used according to the needs of the user. In other words, the user can tailor the system's operation to minimize the execution time or maximize the accuracy of the Edge-Assisted object detector. The exploration of the delay and accuracy trade-off in edge computing has been explored in the past, for example, see [16] and references therein, our work is the first to study this problem in the context of task splitting from servers to end devices. It is noteworthy to point out that this is a new type of offloading architecture, where larger or bigger devices (a server, in this case) splits and outsources their load to many smaller devices (the end nodes, in this case).

This paper follows the following structure: In the next section, we present related works regarding task execution and deep learning application aiming at the optimization of an edge computing system. In Section 3, we present the architecture and system evaluation of our testbed. In Section 4, we focus on the design of equations and task allocation policies that were used to get the results. In Section 5, we demonstrate a discussion with results about the performance of our proposed system. We finally conclude this work in the last section acknowledging limitations and proposing future research directions.

## 2. Related Works

Previous research was presented by Kuanishbay et al. which focuses on optimizing the methods for offloading computations in edge computing networks [17]. More specifically, the authors studied six different optimization methods: convex optimization, Lyapunov optimization, heuristic techniques, machine learning, and game theory method. For each of these methods, the authors focused on the objective functions (minimizing energy consumption, minimizing latency, and minimizing the system utility and the system cost), the application areas, for example, MEC or MCC, the types of offloading methods, and the evaluation methods (simulation or real-world dataset). To conclude, this paper provided a summary of some optimization methods for offloading computations in edge computing networks which can support scholars in their computational offloading research.

A comprehensive survey that provides a thorough examination of computational offloading in MEC, covering various objectives such as minimizing delay, reducing energy consumption, maximizing revenue, maximizing system utility, applications, and offloading approaches, was presented by Chuan Feng et al. [18]. In more detail, the authors conducted a thorough review of the benefits of computation offloading techniques in MEC networks. They analyzed existing works from various perspectives to categorize them based on the application offloading objectives. The authors provide an analytical comparison of various computation offloading techniques, including mathematical solvers, heuristic algorithms, Lyapunov optimization, game theory, the Markov Decision Process, and Reinforcement Learning. They discuss the advantages and disadvantages of each method. Additionally, the paper concludes by highlighting the different challenges faced in computation offloading within MEC networks which must be studied and researched.

Some authors like Wang Xiaofei et al. [19] have also provided a comprehensive survey of edge computing and deep learning. This survey first discusses the fundamentals of this technology, giving some paradigms of edge computing. The concepts discussed are the Cloudlet and micro data centers, mobile edge computing (MEC), fog computing, and collaborative end-edge-cloud-computing. In addition, potential hardware chips for edge AI applications are described. Lastly, the researchers discuss the challenges of the possibility of DL applications being executed on edge devices, focusing on inference, training, and optimization. The contribution of this paper is the provision of insights on how to identify the most suitable edge computing architecture that can optimize deep learning performance for both training and inference and also address challenges like energy consumption, latency, and network communications.

Another promising line of research was introduced by Yuanming Shi et al. [20] which emphasizes the difficulties and solutions of communications in edge AI. The main information that this work is providing is a summary of the communications algorithms for distributed training of various AI models that can be deployed on edge nodes, such as zeroth-order, first-order, second-order, and federated optimization algorithms. Additionally, the authors aim to classify different system architectures of an edge AI system, including partition-based and model partition-based edge training systems. In addition, this paper reviews works based on computation offloading and inferencing at the edge of the network. Finally, different edge AI systems were introduced with discussions focusing on the challenges and solutions of the communications in these systems.

The implementation of deep learning algorithms in edge computing has garnered increasing attention in recent years. Jiasi Chen et al. in [21] presented an analytical review that provides an overview of deep learning applications like computer vision, natural language processing, augmented reality, etc. which are used in edge computing. This work firstly introduces the benefits of bringing the data processing closer to the network edge in other words to the edge nodes or the edge devices. Then the authors provide context for measuring the performance of deep learning models and outline some frameworks for training and inference. The overall purpose of this article was to introduce edge computing and deep learning and present methods for accelerating and evaluating a deep learning inference on edge devices.

Earlier studies also focused on object recognition applications at the network edge. Chu-Fu et al. [22] suggested a cooperative framework for image object recognition, which integrates the process of AIoT devices. They also formulated a task-offloading network optimization problem for this environment and solved it using a heuristic algorithm based on PSO. The testbed that was used in this paper was a practical surveillance system for monitoring human flow, and a technique for detecting error data were also introduced. The encouraging results of the proposed approach could be applied in a real-world surveillance system.

Scholars also discuss how edge computing technology would improve the performance of IoT. Wei Yu et al. in [23] conducted an analytical survey to investigate the relationship between edge computing and the Internet of Things, and how the integration of edge computing can enhance the overall performance of IoT networks. More specifically, this study lists the benefits and divides various edge computing architectures into several categories. In terms of response time computation, capacity, and storage space, it also compares these categories' performance analytically. The performance of IoT devices in the cloud versus edge computing was also compared, and an analytical analysis of the edge computing architecture's performance, task scheduling, security, and privacy was presented. The study outlined the overbalance of edge computing concerning transmission, storage, and computation and provides a discussion about new challenges such as system integration resource management security, and privacy.

Additional studies to understand more completely the key differences between AI for Edge and AI on Edge were introduced by Shuiguang Deng et al. [24]. The authors defined AI for Edge as a potential research direction for giving solutions in terms of optimization problems in edge computing. On the other hand, AI on Edge is a research direction on studying how to run an AI model on Edge. In other words, it is a framework that has been designed to carry out both training and inference operations, with the aim of achieving a particular objective such as optimal algorithm performance, cost savings, privacy, etc. Moreover, in this paper, the authors categorize research efforts in edge computing to certain topologies, content, and services and introduce how to initiate an edge computing application with intelligence.

Researchers also proposed an intelligent and cooperative approach to edge computing in IoT networks, which seeks to achieve a mutually beneficial integration of AI and edge computing [25]. The authors' focus was on redesigning the AI-related modules of edge computing to enable the distribution of AI's core functions from the cloud to the edge. In more detail, the authors introduced a new approach to intelligent and cooperative edge computing in an IoT system, which enables edge computing and AI collaboration. The authors presented different scenarios for their proposed approach. In the first scenario, the edge-AI enables collaboratively efforts between the cloud and the edge. This is accomplished through the implementation of a novel data schema for storage where data are kept locally to ensure privacy. The additional scenario builds a smart edge by leveraging locally adaptive AI. Reservation-based approach for computation offloading that is able to adapt to AI predictions of time sequences. The simulation results indicated that this method not only allowed for the execution of the complete AI model (rather than a simplified version), but also achieved significantly lower latency and higher caching ratios at the edge. These results imply that establishing an edge computing ecosystem as a component of an intelligent Internet of Things system based on services and interfaces helpful to the developers is viable.

In addition, Zhou Zhi et al. [26] have published on edge intelligence. In more detail, this work is offering the background of edge AI, in other words, the architectures, frameworks, and deep learning technologies that are used at the network edge. This paper introduces the basic concepts of AI focusing on deep learning, which is the most popular sector of AI, but also describes the motivation and gives definitions of edge intelligence. This work is essentially an analytical review of the architectures, systems, and framework

for training edge intelligence models that concludes by identifying future directions and challenges of edge intelligence.

In 2019, a framework named Edgent, which utilizes device-edge synergy for collaborative inference in deep neural networks (DNN), with the aim of achieving high inference accuracy and low latency was developed by En Li et al. [27]. The authors after developing this framework implemented it on a desktop PC and a Raspberry Pi and evaluated it based on real work data, and pointed out the efficacy of the Edgent framework. This framework has the ability to compress existing models to accelerate the DNN inference.

To address the issue of task offloading, Firdose Saeik et al. offered a thorough examination of how the edge and cloud may join their computational resources [28]. Its discussions revolve around artificial intelligence, mathematical, and control theory optimization approaches. These can be utilized to meet diverse constraints, limitations, and dynamic problems pertaining to end-to-end application execution. Unlike other studies, this work on task offloading contributes in a twin way. Firstly, it presents a survey on task offloading that covers three sub-fields, namely optimization algorithms, artificial intelligence techniques, and control theory. Secondly, it categorizes the discussed techniques based on various factors, such as the objective functions, granularity level, utilization of edge and cloud infrastructures, and incorporation of mobility depending on the type of edge devices.

As far as collaborative task execution goes, Galanopoulos et al. developed a framework that enables a collaborative task execution on edge devices [29]. This task execution is based on an auction algorithm that aims to optimize the accuracy and the execution delay. The author of this paper implemented and evaluated the system performance in relation to accuracy and execution delay by implementing a facial recognition application on a Raspberry Pi. The proposed algorithm outperforms several benchmark policies.

Among many object recognition models, one of the most popular is the YOLO network which can implement a real-time application at the network edge. Haogang Feng et al. [30] presented a benchmark evaluation and analysis of the YOLO object detector on edge class devices. In this work the YOLO object detector was implemented with an NVIDIA Jetson Xavier, an NVIDIA Jetson Nano, and a Raspberry Pi 4 in conjunction with a USB neural network compute processor. Four versions of the YOLO object detector were tested on the three edge devices using videos with different input-size windows. The findings indicated that the Jetson Nano achieves optimal performance compared to the other two edge devices in terms of high performance and cost. In particular, the Jetson Nano gets 15 FPS while running YOLOv4 tiny model.

A recent study by Galanopoulos et al. developed an object detection system at the network edge that identifies the system-level trade-offs between E2E latency and execution accuracy [16]. The authors developed methods that optimize the system's transmission delay and highlight the impact of the image encoding rate and size of the neural network. Their results were based on a real-time object recognition application. They showed that the image level compression as well as their neural network size are key parameters that affect object recognition accuracy and also latency from one end to the other.

A mobile augmented reality for real-time object detection for was introduced by Luyang Liu et al. [31]. This system can achieve accurate object detection for commodity AR/MR with a frame rate of 60 fps. The authors of this work demonstrated that using low-latency offloading techniques can effectively decrease the offloading latency and the amount of bandwidth consumed. The authors prototyped an E2E system that is implemented and used on mobile devices. The results showed that the system after using smart offloading techniques and fast object detection methods to sustain the detection accuracy can reduce the false detection rage by 27–38% and increase the detection rate by 20–34%.

Cheng Dai et al. [32] presented an enhanced object detection system for an edge-based Internet of Vehicles (IoV) system, aiming to reduce latency. They proposed a new method for object detection based on a video key-frame, which can achieve a compression ratio of 60% with an error rate of only 10%. The proposed method can also accelerate the processing of frames in the edge IoV system.

Finally, Dong-Jin-Shin et al. [33] presented an analytic deep-learning performance evaluation using the YOLO object detector in the NVIDIA Jetson platform. The authors used general deep learning frameworks like TensorRT and TensorFlow-Lite on embedded and mobile devices to get an analytic performance evaluation. The evaluation was performed on the NVIDIA Jetson AGX Xavier platform employing CPU and GPU utilisation, latency, energy consumption, and accuracy. The evaluation model was the YOLOv4 object detector using COCO, PASCAL, and VOC Datasets. The findings indicated that the implementation of a deep learning model on a mobile device using the TensorFlow-Lite framework is the best solution. On the contrary, when implementing a deep learning model on an embedded system with a built-in Tensor core the most effective framework is the TensorRT framework.

All in all, the studied literature provides evidence that the processing of the computing workloads is flowing from the cloud to the edge and that AI workload in edge computing architectures is a rapidly emerging research field. Moreover, a great effort has been made by the research community in designing optimization algorithms to improve the performance of edge computing systems. The most common solution in task offloading approaches is that the workload is transferred to local servers which are making the computation. Our work is, as far as we know, the first study that explores the problem of executing the workloads by splitting the task from servers to end devices.

## 3. Architecture and System Evaluation

### 3.1. Hardware and Software Setup

We propose an edge computing topology that consists of an edge server, a Wi-Fi access point, and three end devices. The end devices are three Raspberry Pi's 4 B models, two with 4 GB RAM and one with 2 GB RAM. The first Raspberry Pi with 2 GB RAM has a Google Coral TPU accelerator as an inference machine and the other two have a CPU-based TensorFlow-Lite inference machine. The edge server software is written in Python and it is responsible for offloading object detection tasks to the three end devices based on the needs of the user. Specifically, the server sends sets of images to the end devices to apply an object detector on the transmitted images. The end devices send the results (i.e., labels for every processed image and the bounding boxes) back to the server. The object detection is performed by YOLOv5 [15], which is a deep learning object detector [30]. The YOLO model takes as input an n x n array of image pixels. Each pixel is a float or int value, which down-samples the array by a specific number to give a grid. These grid cells propose the labels and the bounding boxes for every object contained in the dataset. As a result, a set of bounding boxes of the recognised objects along with their labels and their confidence values are generated. The proposed edge computing topology consists of an edge server which is a laptop with an i7-6700HQ CPU processor with 8 GB RAM, the three Raspberry Pi's described above, and a wireless access point which are shown in Figure 1.

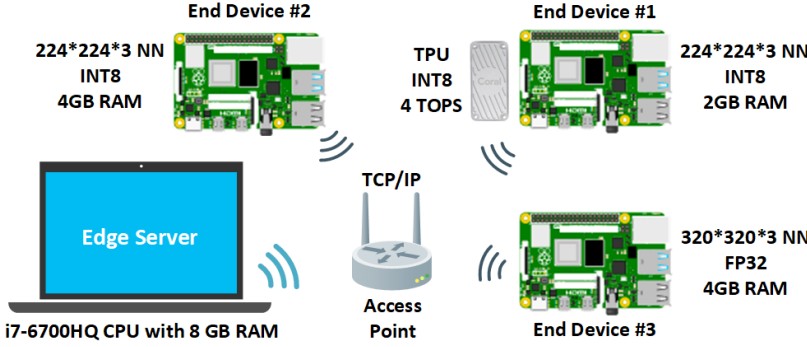

**Figure 1.** An illustration of our testbed.

### 3.2. The Need for Edge Server Offloading

Our work looks into the concept of offloading the processing of the workload optimally on multiple end devices to achieve faster processing, or high-accuracy object detection of the images. In other words, we tried to utilize all the processing resources that are close to the server to achieve all the benefits provided by edge computing technology. We used the YOLOv5 version of the object detector, which can be easily implemented into Raspberry Pi along with different task-splitting strategies and compression decisions, which attempt to offload the workload to the end devices optimally. The offloading of the workload is done by the edge server using TCP Sockets. We used this API (Sockets) for network communication to ensure reliable data transfers.

### 3.3. Measurements of the Different Delay Components

Our first priority was to measure all the systems' E2E latency components and see how the encoding rate $q$ and network status impact them. The E2E latency when processing the set of images is determined by the slowest subtask execution. Namely, when we partition the image set, we essentially create different subtasks and we assign each of them to a different end device. The completion of each subtask requires the respective images to be sent to that device, processed there, and the results (bounding objects and labels) to be returned to the server. Given the different hardware specifications of the end devices and the volatility of the wireless channels, these delays are challenging to predict or capture analytically as a function of the splitting and compression decisions. Our goal was to find the right encoding rate $q$ in which the image set sizes are as small as possible to minimize the transmission delay or more specifically the time needed to offload the images to the end devices and affect the performance of the YOLO model on every end device as less as possible.

#### 3.3.1. Encoding Delay

The edge server makes the JPEG compression before the transmission (offloading) to the end devices. Image encoding is a very important step in our system because it makes the size of the image sets much smaller which has an enormous effect on the transmission delay. The compression of the image sets is made by the encoding rate value $q$ [10,95]. The higher the encoding rate value the bigger the image set sizes. Figures 2–5 present how the encoding rate affects the encoding delay and how much the encoding delay affects the image set sizes.

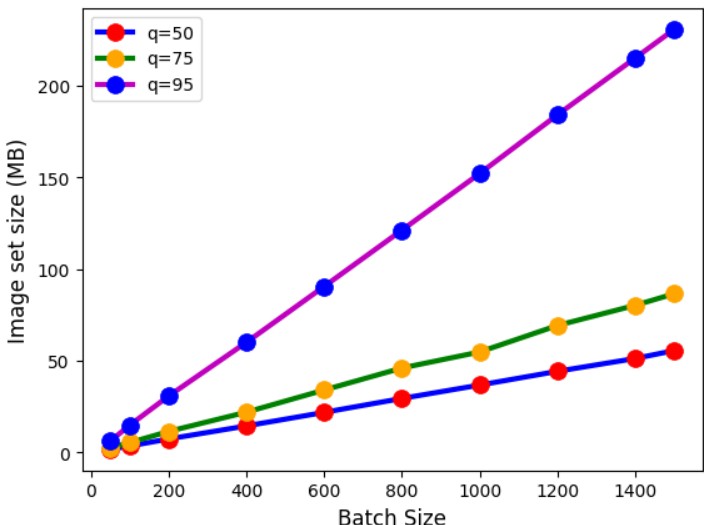

**Figure 2.** Image size for different batch sizes (1500 COCO images) for encoding rate $q$ = [50,75,95].

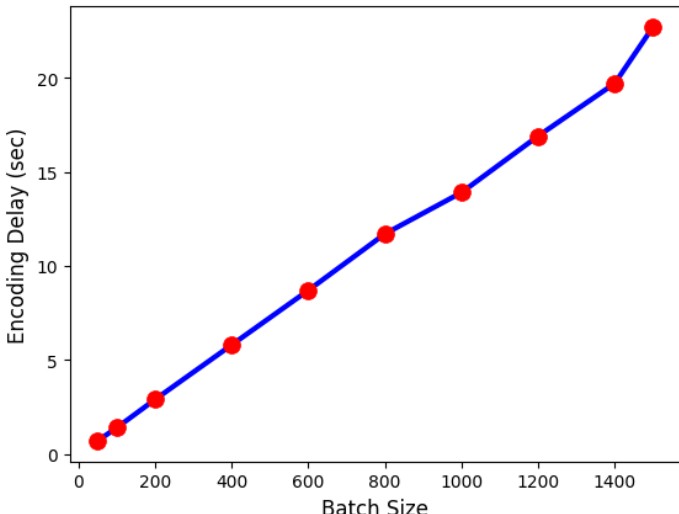

**Figure 3.** Encoding delay for every batch size ($q$ = 50).

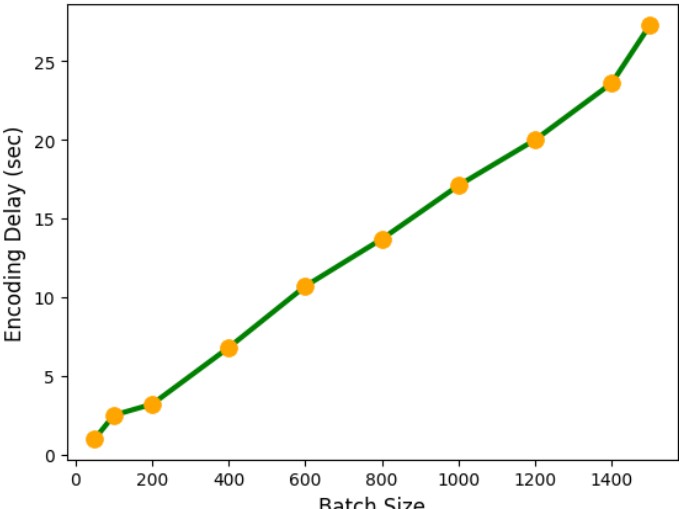

**Figure 4.** Encoding delay for every batch size ($q$ = 75).

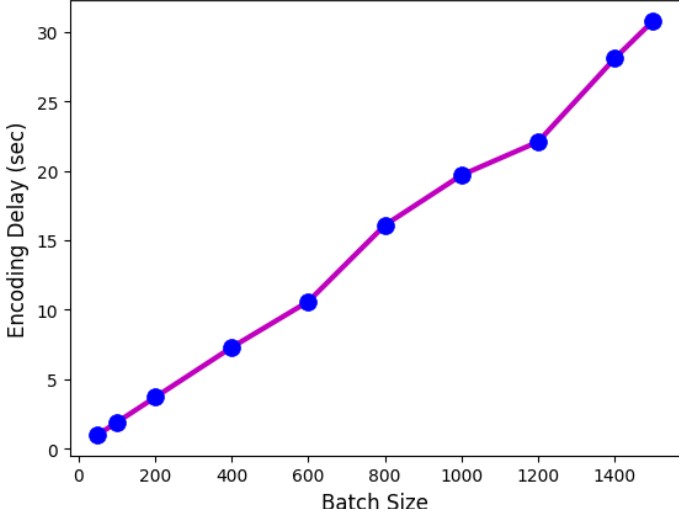

**Figure 5.** Encoding delay for every batch size ($q$ = 95).

3.3.2. Transmission Delay

Next, we presented the impact of the channel conditions and the encoding rate on the transmission delay of different image sets (batch sizes). The size of the transmitted image sets was between 6–230 MB. Figures 6–8 present how the transmission delay is affected by the encoding rate value. Our measurements were performed with three different channel conditions. The conditions of the three channels as shown in Figures 6–8 are good, medium, and bad channel conditions respectively. As expected the higher the encoding rate value $q$, the bigger the transmission delay. As mentioned before we proposed to develop a technique in which we minimize the transmission delay by sacrificing the least possible accuracy of the Edge-Assisted object detection system. In more detail, we suggested a method to minimize transmission delay while maintaining high accuracy in the Edge-Assisted object detection system. Through our testing, we have determined that the most effective value for $q$, which achieves minimal transmission delay without compromising system accuracy is $q = 50$ which is clearly shown in Table 1. The accuracy is affected by the compression in a non-deterministic, and a priori unknown, fashion. Indeed, as prior studies have shown, the detection accuracy depends on how well the training data match the test data, the efficacy of the particular ML library we use, and several other latent factors. Hence, whether the compression affects significantly the accuracy on top of all these factors, or whether it is less important, is something that we cannot assess in advance, and it is not realistic to assume that we have an analytical expression (i.e., function) for its effect.

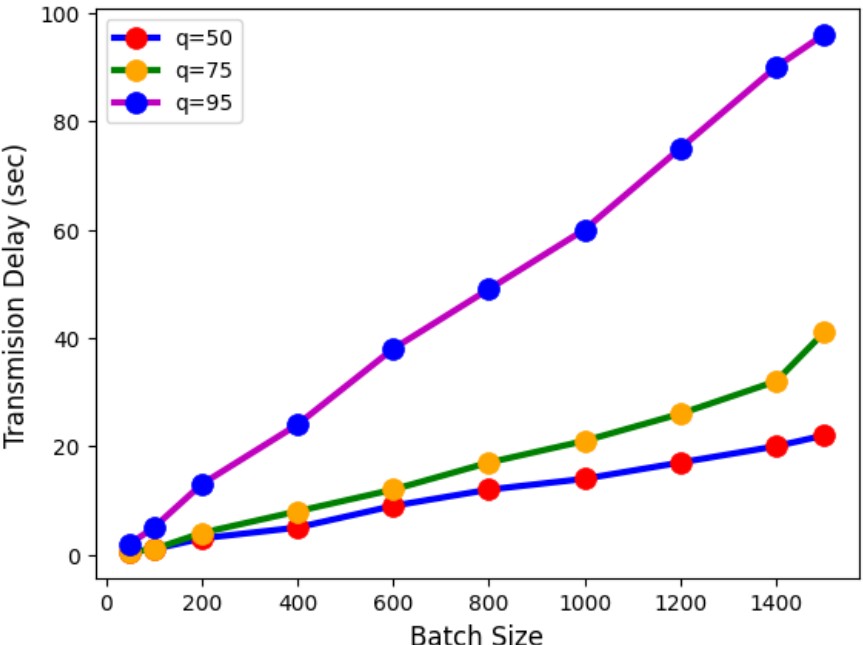

**Figure 6.** Transmission delay for different batch sizes with good channel conditions.

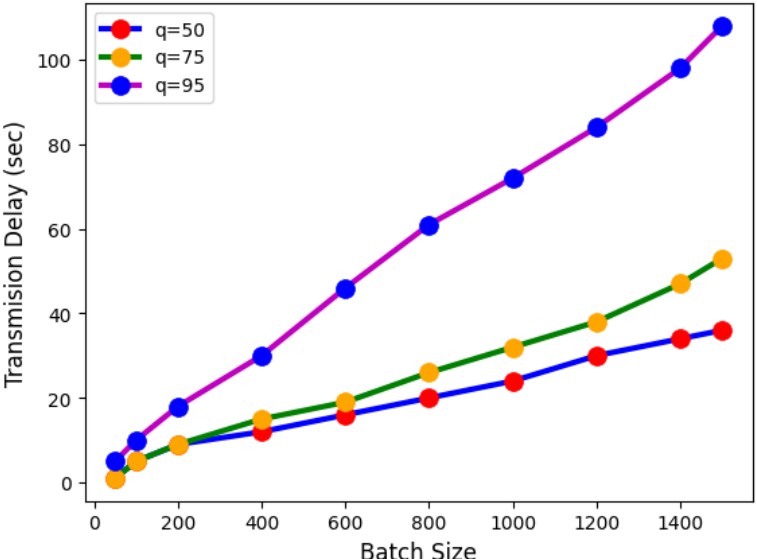

**Figure 7.** Transmission delay for different batch sizes with medium channel conditions.

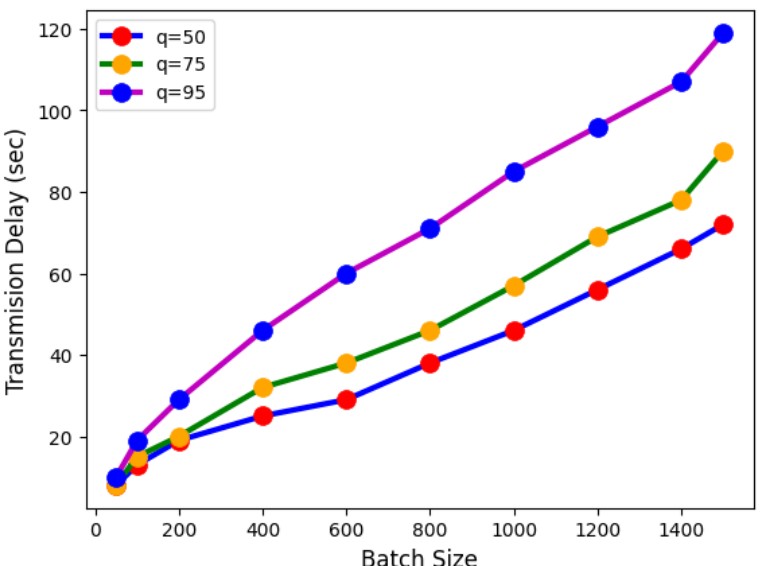

**Figure 8.** Transmission delay for different batch sizes with bad channel conditions.

*3.4. Evaluation Scenario*

For the evaluation of the system's performance, we used the COCO dataset [34] which consists of a big collection of images and objects with corresponding ground truth annotations. For the evaluation of the models performance, we used the precision, recall, and mean average precision metrics for a range of intersection over unit values (IoU). The detection was considered successful as a true positive if the intersection of the unit value was bigger than 0.5. The edge server included all of the COCO images data, and using a Python script, the server sent subsets of those data to the end devices. We used 1500 images from this validation set; images were divided into subsets of images and sent for processing to different end devices, with different hardware and/or inference machines, and different network conditions. This allows us to have a better understanding of the system's operation in diverse scenarios. Apart from revealing the trade-offs and challenges of this problem, our results can also be us to create look-up tables with the best configurations for different systems and cases. Furthermore, for the evaluation of the system's performance, we used metrics that measure the time needed to process every subset; in other words inference

processing. We also used metrics that measure the CPU utilization for the processing of every image subset [33] as shown in Section 3.6.

### 3.5. Inference Metrics

Metrics for measuring machine learning inference performance are presented in recent works [35,36]. In our study, the following metrics were used to conduct our experiment:

- **Transmission delay:** is the length of time needed to transfer the images to a destination device.
- **True Positive (TP):** In the context of image classification, a true positive is when the model correctly predicts that an image belongs to a certain class, and the image actually belongs to that class.
- **True Negative (TN):** A true negative is when the model correctly predicts that an image does not belong to a certain class, and the image actually does not belong to that class.
- **False Positive (FP):** A false positive is when the model incorrectly predicts that an image belongs to a certain class, when in fact it does not.
- **False Negative (FN):** A false negative is when the model incorrectly predicts that an image does not belong to a certain class, when in fact it does.
- **Precision:** is the fraction of total True positives predictions of our model divided by the sum of True Positive and False Positive predictions, as presented in the following equation:

$$Precision = \frac{TP}{TP + FP} \tag{1}$$

- **Recall:** is the fraction of total True positives predictions of our model divided by the sum of True Positive and False Negative predictions, as presented in the following equation:

$$Recall = \frac{TP}{TP + FN} \tag{2}$$

- **Mean Average Precision (mAP):** is a metric used in object detection tasks to evaluate the model's performance. More specifically, mAP is calculated by finding the Average Precision (AP) for each class ($i$) and then the average over a number of classes. N is the total number of classes. The accuracy of the object detector is obtained by this metric:

$$mAP = \frac{1}{N} \sum_{i=1}^{n} AP_i \tag{3}$$

- **Execution Time:** is the time for processing every image set.

### 3.6. Comparative Analysis and Evaluation of the YOLOv5 Inference Using the COCO Dataset

As mentioned Table 1 presents map values for all $q$ values used. Tables 1–4 present the results of measuring the YOLOv5 inference performance through 15 image sets taken from the validation set of the COCO Dataset [34]. Every image set consists of 100 images. The used YOLO models have different input image sizes and precisions. Moreover, they are implemented into various end devices with different inference machines to reveal sharp trade-offs. Firstly, in Table 2, we implemented the YOLO model with input image size $224 \times 224 \times 3$ and INT8 precision into a Raspberry Pi 4 with the TensorFlow-Lite inference machine. Secondly, in Table 3, the Raspberry Pi 4 with 2 GB RAM and the google coral TPU using the edge TPU inference machine have also implemented the YOLO model with input image size $224 \times 224 \times 3$ and INT8 precision. Finally, in Table 4, the last end device has implemented the YOLO model with input image size $320 \times 320 \times 3$ and float32 precision. Comparing the results from the three tables we observe that the edge TPU inference machine executes the same workload almost 8 times faster than the TensorFlow-Lite inference machine with NN input size $320 \times 320 \times 3$ and almost 2 times faster than the TensorFlow-Lite inference machine with NN input size $224 \times 224 \times 3$.

**Table 1.** Map values for all $q$ values used (1500 COCO images).

| Inference Machine | NN Size | mAP for $q = 50$ | mAP for $q = 75$ | mAP for $q = 95$ |
|---|---|---|---|---|
| TensorFlow-Lite | $224 \times 224 \times 3$ | 0.46 | 0.46 | 0.47 |
| TensorFlow-Lite | $320 \times 320 \times 3$ | 0.54 | 0.55 | 0.55 |
| Edge-TPU | $224 \times 224 \times 3$ | 0.44 | 0.44 | 0.45 |

**Table 2.** Evaluation of the YOLOv5 with TensorFlow-Lite Inference using the COCO Dataset and NN size $224 \times 224 \times 3$.

| Image Set | Execution Time (s) | Average CPU Utilization (%) | Precision | Recall | mAP |
|---|---|---|---|---|---|
| Image Set1 | 23.3 | 27.5 | 0.67 | 0.42 | 0.47 |
| Image Set2 | 23.6 | 27.9 | 0.69 | 0.34 | 0.41 |
| Image Set3 | 23.5 | 28.1 | 0.66 | 0.40 | 0.47 |
| Image Set4 | 23.1 | 26.8 | 0.57 | 0.38 | 0.41 |
| Image Set5 | 23.0 | 24.8 | 0.74 | 0.44 | 0.52 |
| Image Set6 | 23.4 | 26.2 | 0.55 | 0.42 | 0.45 |
| Image Set7 | 23.3 | 26.5 | 0.61 | 0.42 | 0.46 |
| Image Set8 | 24.0 | 27.2 | 0.67 | 0.40 | 0.46 |
| Image Set9 | 23.0 | 28.2 | 0.49 | 0.44 | 0.45 |
| Image Set10 | 23.2 | 28.9 | 0.57 | 0.40 | 0.45 |
| Image Set11 | 23.5 | 27.2 | 0.60 | 0.35 | 0.41 |
| Image Set12 | 23.1 | 27.4 | 0.55 | 0.38 | 0.42 |
| Image Set13 | 23.3 | 27.7 | 0.59 | 0.38 | 0.44 |
| Image Set14 | 23.3 | 27.5 | 0.57 | 0.36 | 0.43 |
| Image Set15 | 23.1 | 27.8 | 0.61 | 0.39 | 0.44 |

**Table 3.** Evaluation of the YOLOv5 with Edge TPU Inference Machine using the COCO Dataset and NN size $224 \times 224 \times 3$.

| Image Set | Execution Time (s) | Average CPU Utilization (%) | Precision | Recall | mAP |
|---|---|---|---|---|---|
| Image Set1 | 9.6 | 29.8 | 0.66 | 0.41 | 0.46 |
| Image Set2 | 9.0 | 33.2 | 0.54 | 0.36 | 0.40 |
| Image Set3 | 9.9 | 32.8 | 0.65 | 0.46 | 0.47 |
| Image Set4 | 9.1 | 32.2 | 0.53 | 0.38 | 0.41 |
| Image Set5 | 9.4 | 34.2 | 0.58 | 0.52 | 0.55 |
| Image Set6 | 9.4 | 33.6 | 0.71 | 0.38 | 0.44 |
| Image Set7 | 9.0 | 34.1 | 0.67 | 0.39 | 0.45 |
| Image Set8 | 9.8 | 32.5 | 0.58 | 0.43 | 0.45 |
| Image Set9 | 9.2 | 31.5 | 0.63 | 0.38 | 0.45 |
| Image Set10 | 9.4 | 32.0 | 0.70 | 0.37 | 0.48 |
| Image Set11 | 9.3 | 30.9 | 0.71 | 0.35 | 0.41 |
| Image Set12 | 8.7 | 31.2 | 0.54 | 0.40 | 0.42 |
| Image Set13 | 8.9 | 32.5 | 0.56 | 0.43 | 0.43 |
| Image Set14 | 9.2 | 30.9 | 0.59 | 0.35 | 0.40 |
| Image Set15 | 9.3 | 34.2 | 0.58 | 0.42 | 0.45 |

**Table 4.** Evaluation of the YOLOv5 with TensorFlow-Lite Inference Machine using the COCO Dataset and NN size 320 × 320 × 3.

| Image Set | Execution Time (s) | Average CPU Utilization (%) | Precision | Recall | mAP |
|---|---|---|---|---|---|
| Image Set1 | 83.3 | 27.1 | 0.71 | 0.53 | 0.6 |
| Image Set2 | 81.1 | 29.3 | 0.77 | 0.43 | 0.56 |
| Image Set3 | 82.1 | 29.2 | 0.73 | 0.50 | 0.54 |
| Image Set4 | 83.6 | 28.1 | 0.6 | 0.49 | 0.55 |
| Image Set5 | 83.4 | 28.5 | 0.71 | 0.55 | 0.62 |
| Image Set6 | 81.3 | 29.6 | 0.6 | 0.48 | 0.52 |
| Image Set7 | 81.6 | 29.8 | 0.76 | 0.5 | 0.55 |
| Image Set8 | 81.7 | 28.8 | 0.75 | 0.5 | 0.56 |
| Image Set9 | 82.6 | 29.6 | 0.47 | 0.55 | 0.55 |
| Image Set10 | 83.1 | 29.3 | 0.54 | 0.53 | 0.54 |
| Image Set11 | 82.1 | 28.4 | 0.62 | 0.48 | 0.51 |
| Image Set12 | 82.8 | 30.3 | 0.67 | 0.47 | 0.51 |
| Image Set13 | 81.1 | 28.5 | 0.6 | 0.49 | 0.50 |
| Image Set14 | 81.4 | 27.2 | 0.57 | 0.49 | 0.51 |
| Image Set15 | 81.5 | 27.8 | 0.62 | 0.52 | 0.58 |

## 4. Design of Algorithms and Decision Variables

The goal of our experiments was to develop an edge-assisted object detection system that is tailored around the needs of the user to minimize the E2E execution time or to maximize the accuracy of the system. Hence, in this paper, we follow an experimental approach where we use our testbed to evaluate an exhaustive list of possible configurations, i.e., different splitting decisions and compression values, with respect to accuracy and delay criteria. This analysis sheds light on the dependencies and conflicts of these two criteria and allows one to select, offline, a strategy that optimizes the criterion of their preference. To make this clear, we also introduce the respective optimization problems, albeit without providing analytical expressions for the metrics, for the reasons we explained above. The aim of this research was to find the optimal trade-off of the metrics to improve E2E execution time and the accuracy of the system. The key decisions here are how to split and offload the set of images, across the end devices. This decision is taken at the server, based on the solution of our properly defined optimization problems. In the experiments, we demonstrated that the parameters which affect and improve the operation of the system are the encoding rate $q$ (compression value) and the batch sizes (splitting decisions) of the delivered images to every device. More specifically based on our measurements (Tables 1–4 and Figures 2–10) we argue that the parameters which affect the accuracy and execution time are: the encoding delay $T_{enc}$, the transmission delay $T_{tx}$, the inference delay $T_{inf}$ and $T_{dl}$ depend on decision variables $q$ and the percentage of the total amount of images transmitted to each device $a_i$. Therefore, the decision variables of our system are the encoding rate $q$ and $a_1$, $a_2$, $a_3$.

It should be noted that we define our functions and our parameters in Table 5; these parameters were used for constructing our optimization problem. In more detail, we define:

$$\sum_{i=1}^{3} a_i = 1 \qquad with \quad a_i \in [0,1] \tag{4}$$

$$b_i = a_i * L \tag{5}$$

$$T_{dl}(q, a_i) = max([T_{enc}(q) + T_{tx}(q) + T_{inf}(a_1);$$
$$T_{enc}(q) + T_{tx}(q) + T_{inf}(a_2); \qquad (6)$$
$$T_{enc}(q) + T_{tx}(q) + T_{inf}(a_3);)])$$

**Table 5.** Definitions of variables and functions.

| Description | Parameter |
|---|---|
| Total number of images | L |
| Percentage of the total number of images for every device i | $a_i$ |
| Number of images sent for processing to every device i | $b_i$ |
| Inference machine of end device i | $i_{nfi}$ |
| Time to process the images send | $T_{inf}$ |
| Time to encode the images | $T_{enc}$ |
| Time to transmit the images | $T_{tx}$ |
| Time to process all the images for the three devices | $T_{dl}$ |
| Total execution time | $T_{total}$ |
| Systems execution accuracy | $S_{acc}$ |

In Equation (6), we calculate the three values of time from which we keep the largest value. More specifically offloading to every device in a workload means that we calculate three values of time from which we keep the time value from the device which ended the processing of the images last. In this work, we formulate two research questions or optimization problems. The first one refers to the case *Q1* where we minimize the E2E execution time. The second *Q2* refers to the case, where we maximize accuracy. The optimization problems can be written as:

$$Q1: \quad minimize \quad T_{total}(q, a_i) \qquad (7)$$

$$Q2: \quad maximize \quad S_{acc}(q, a_i) \qquad (8)$$
$$and \quad T_{total} \geqslant T_{min}$$

from Equations (4)–(6) we define:

$$T_{total}(q, a_i) = T_{dl}(q, a_i) \qquad (9)$$

Taking into account our evaluation results of the YOLOv5 using the COCO dataset in Tables 1–4, and also Figures 9 and 10, we reveal that the edge TPU inference machine makes the processing of the workload faster than the other two inference machines, so the common logic says to send most of the images to the edge TPU inference and lesser images to the other two inference machines to answer to research question *Q1*. On the other hand, to answer *Q2* we have to offload the specific batches of images to get maximized accuracy.

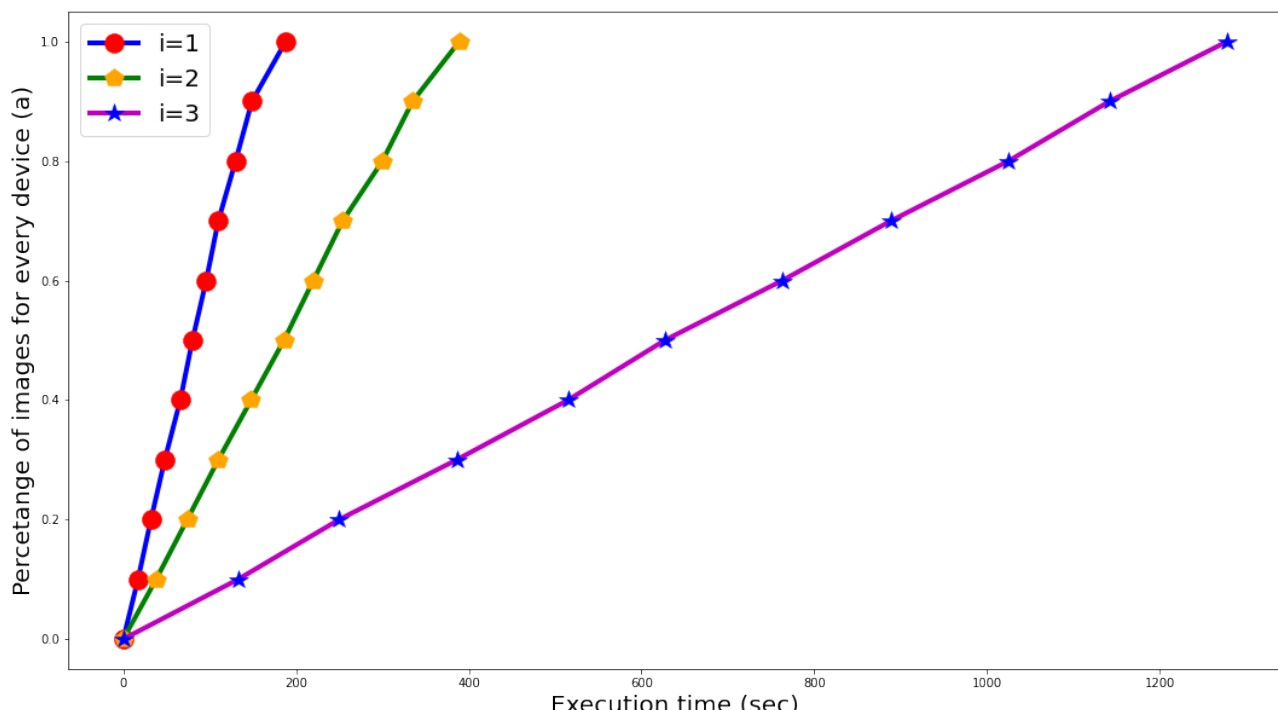

**Figure 9.** E2E execution time values for every device and batch size.

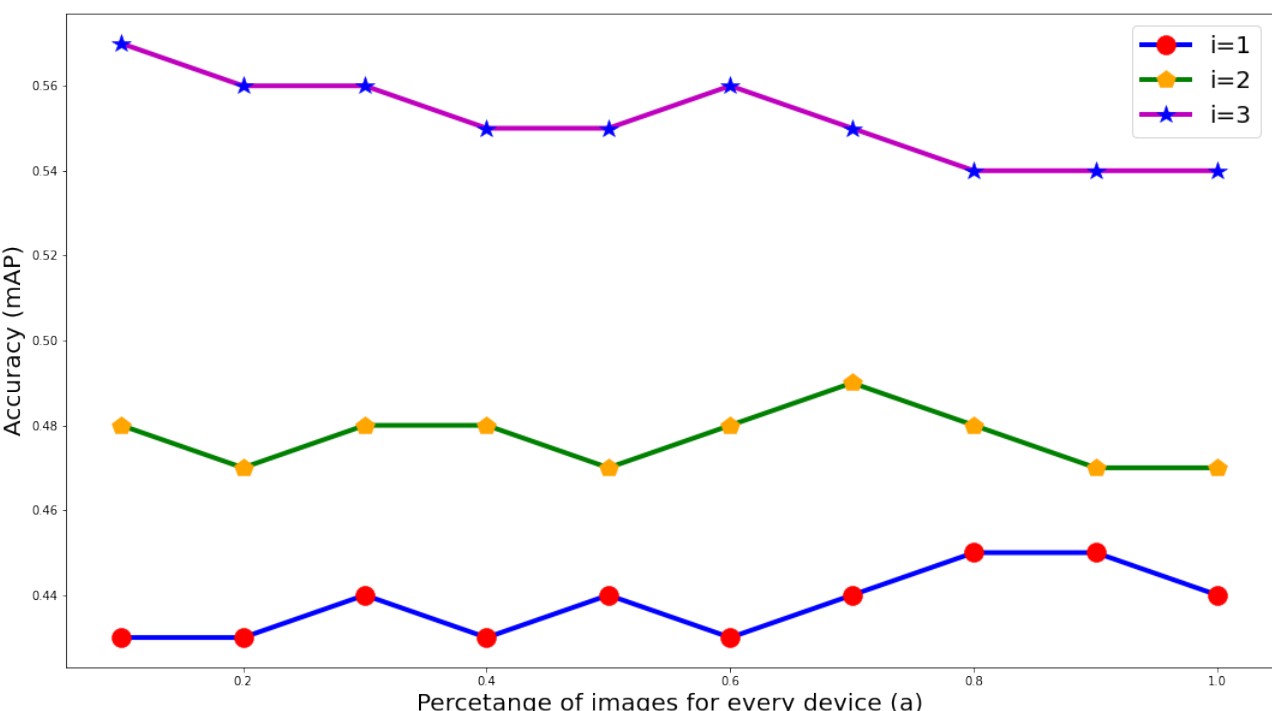

**Figure 10.** Accuracy values for every device and batch size.

## 5. Results and Discussion

In this section, we present our results that solve the two optimization problems "manually" by finding the optimal values of the decision variables $a_i$ and $q$. We define for i = 1 the edge TPU inference machine, i = 2 the TensorFlow-Lite inference machine with $224 \times 224 \times 3$ NN input size, and i = 3 the TensorFlow-Lite inference machine with $320 \times 320 \times 3$ NN input size. $T_{min}$ is the target execution time requested by the user.

In Figure 11, we observe that the Pareto plot depicts the most efficient solutions for our two optimization problems. Every solution in the Pareto plot combines three $a_i$ values. Regarding the optimal value for the encoding rate, it is clearly shown in Figures 6–8 that the optimal value for the encoding rate is $q = 50$. To answer *Q1* according to the Pareto, the optimal solution to get the minimum E2E execution time is $T_{total} = 110$ s. The most optimal values for $a_i$ are $a_1 = 0.7$, $a_2 = 0.3$, $a_3 = 0$. For *Q2*, we define it as $T_{min} = 400$ (s), which means that all the feasible solutions can not be greater than 400 s. According also to the Pareto plot, the most optimal solution for *Q2* achieves $S_{acc} = 0.53$ and $T_{total} = 342$ s. The values for $a_i$ which achieve the maximum accuracy $a_1 = 0$, $a_2 = 0.8$, $a_3 = 0.2$.

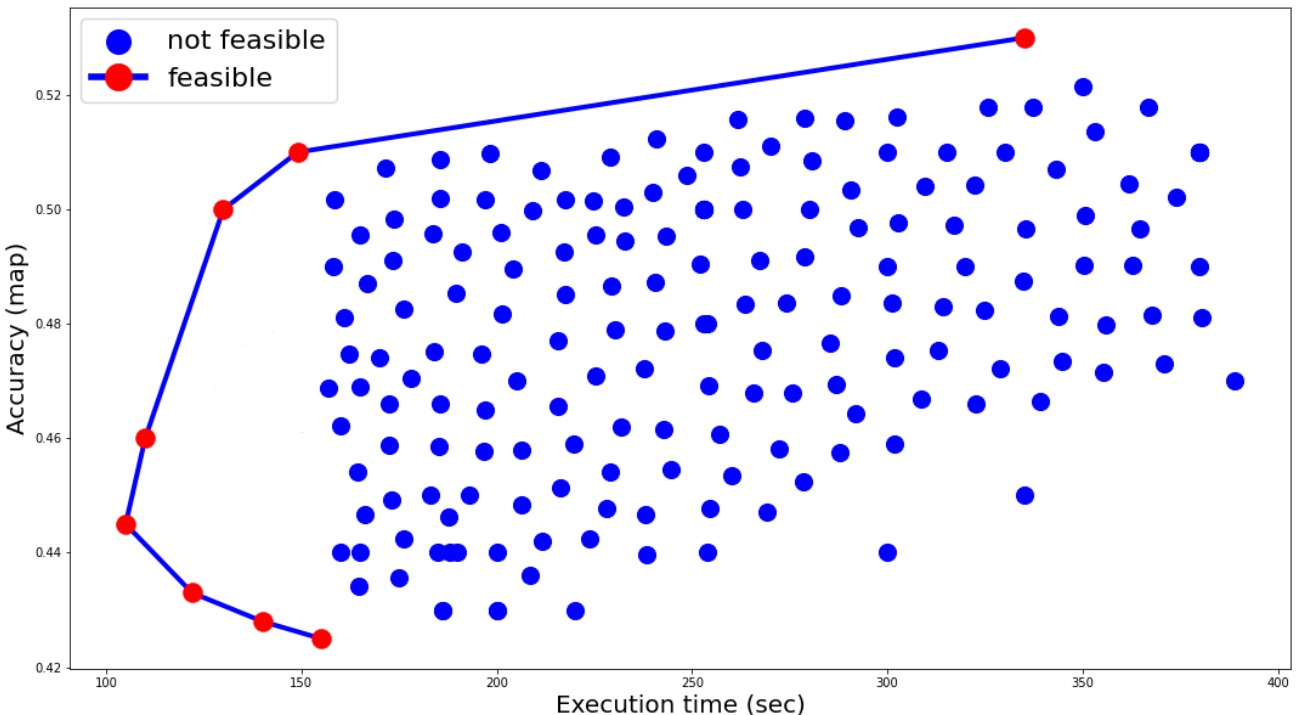

**Figure 11.** Most best solutions (Pareto front) for *Q1*, *Q2*.

The main idea of this paper was to develop a cooperative task execution of an edge-assisted object detection system. In our edge computing system, we used the You Only Look Once (YOLO) version 5. We first evaluate the performance of the YOLOv5 on every end device using metrics like Precision, Recall, and mAP on the COCO dataset. The goal of our work was to offload the workload (batch of images) to three end devices that are in our testbed and try to get the minimum execution time or the maximum accuracy. We noticed that the parameter-decision variables, which affect the E2E execution time, and execution accuracy are: the encoding rate $q$ (compression value); and the batch sizes (splitting decisions) sent to every device.

More specifically, we explored the optimal trade-offs between the parameters and decision variables to solve optimization problems "manually". We found that the key parameter which affects an edge-assisted object detection system is the encoding rate (compression value) along with the neural network input size and the precision (INT8, float32) and also the batch sizes (splitting decisions) sent to every end device.

Actually, the effectiveness of image compression on object recognition accuracy has recently gained much attention from many researchers. The authors in [37] showed the significance of the effectiveness of image compression or better encoding rate $q$ on the object detection accuracy. The system-level trade-offs between E2E latency and deep learning accuracy were recently introduced in [16]. We tried to learn from these lessons and add value to the relevant literature.

The contribution of our work is that we propose an edge-assisted object detection system that offloads the workload from the edge server to the end devices based on the needs of the user rather than offloading the workload from the end devices to the edge server and the cloud [32,37].

This approach has not been previously explored. In our work, we highlight the optimal trade-offs between E2E execution time and accuracy in an edge computing architecture that other researchers have also studied. Our work, however, goes beyond that by providing a real-time edge-assisted object detection system that is tailored made according to the needs of the user by offloading the workload to the three devices. Our proposed offloading strategies can easily be implemented on end devices, which means that the latency of our system can be reduced much more.

A natural next step in this line of work is to devise algorithms that can select the configuration decisions (splitting and compressing) in a dynamic fashion. Another future research direction associated with our subject of work could be about addressing the challenge of finding more effective image compression methods like autoencoders aiming to get better results in transmission delay and also to study ways to improve the accuracy metric with other Deep Learning techniques like distillation. Therefore, it would be useful to study autoencoders as image compressors and the distillation method and see how these affect the accuracy and execution time in an edge-assisted object detection system.

The limitation we faced in this work was, as said before, that the proposed system cannot make the splitting and compression decisions in a dynamic fashion which means that the user has to take into account all the feasible solutions and choose manually the optimal decision variables the get the minimized execution time or the maximized accuracy.

## 6. Conclusions

In this work, we proposed and developed a cooperative task execution of an edge-assisted object detector system and presented how to optimal deliver (offload) batches of images to minimize the E2E latency or maximize the execution accuracy of the system. Our purpose was to design an edge-assisted object detection that is tailored around the needs of the user to minimize the E2E execution time or to maximize the accuracy. After presenting our two optimization problems to solve them, we followed an experimental approach by testing all the possible configurations of the parameter which affect the system performance (execution time, accuracy). After testing all the possible configurations of the parameters which are presented in the Pareto plot, we found the optimal trade-offs of our parameters, which solve the two optimization problems "manually". It is concluded that the parameters which affect the E2E execution time and accuracy of the system are the encoding rate $q$ (compression value) and the batch sizes (splitting decisions), which are sent to every end device. The key contribution of this study is that we highlight the optimal trade-offs of our parameters, which can be applied according to the needs of the user to minimize E2E latency or maximize the execution accuracy of the system. We demonstrated that offloading the workload on multiple end devices based on different splitting decisions and compression values can improve the system's performance to respond in real-time conditions without the need for a server or cloud resources.

**Author Contributions:** Conceptualization, D.K. and P.A.; methodology, D.K., G.I. and A.N.; software, P.A.; validation, G.I., T.L. and D.K.; formal analysis, D.K., T.L. and P.A.; data curation, P.A.; writing—original draft preparation, P.A. and D.K.; writing—review and editing, G.I., T.L. and A.N.; visualization, D.K. and P.A.; supervision, D.K. All authors have read and agreed to the published version of the manuscript.

**Funding:** This research received no external funding.

**Conflicts of Interest:** The authors declare no conflict of interest.

## Abbreviations

The following abbreviations are used in this manuscript:

| | |
|---|---|
| AI | Artificial Intelligence |
| AIoT | Artificial Internet of Things |
| AR | Augmented Reality |
| COCO | Common Objects in Context |
| CPU | Central Processing Unit |
| E2E | End-to-End |
| EC | Edge Computing |
| FPS | Frames Per Second |
| IoT | Internet of Things |
| IoU | Intersection over Unit |
| IoV | Internet of Vehicles |
| mAP | mean Average Precision |
| MCC | Mobile Cloud Computing |
| MEC | Mobile Edge Computing |
| s | Seconds |
| SBC | Single Board Computer |
| TCP | Transmission Control Protocol |
| TPU | Tensor Processing Unit |
| YOLO | You Only Look Once |

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
