# Peer review of "Cooperative Task Execution for Object Detection in Edge Computing: An Internet of Things Application"

_applsci, doi:10.3390/app13084982_

Round 1

Reviewer 1 Report

Review Report

The authors of this study examine a system for collaborative task execution in an edge computing architecture. They primarily aim to offer optimization for inferencing deep learning models in terms of execution speed and accuracy. Additionally, they emphasise the crucial roles of efficient picture compression and network status as they design strategies to reduce the latency in images being transmitted from the edge server to the end devices. Lastly, they show that spreading out workloads across a number of endpoints enhances the system's ability to react to real-time situations without the use of server or cloud resources.

Hence, I do recommend the paper for possible publication in your reputed journal.

Author Response

Dear Reviewer,

We really want to thank you for providing us the opportunity for a resubmission and for your dedicated effort in helping us to improve the quality of our article respectively.

Reviewer 2 Report

Summary:

1-In this article, a cooperative task execution system in an edge-computing architecture had been investigated.

2-In the proposed topology, the edge server offloads different workloads to end devices, which collaboratively execute object detection on the transmitted sets of images.

3-The proposed system attempts to provide optimization in terms of execution accuracy and execution time for inferencing deep learning models.

4-developing techniques to optimize the transmission delay of the images from the edge server to the end devices had been focused on.  Those techniques had been developed by highlighting the key role of effective image compression and network status.

5-It had been demonstrated that offloading workloads on multiple end devices improve the system’s performance to respond in real-time conditions without the need for the server or cloud resources.

the steps that this study will follow will be:

1-In the proposed heterogeneous testbed, the You Only Look Once (YOLO) version 5 had been used. It is one of the most popular object detectors. An edge server and three different end devices were used with different characteristics like CPU/TPU, different sizes of RAM, and different neural network input sizes to identify sharp trade-offs.

2-First, the performance of the YOLOv5 on the end devices using metrics like Precision, Recall, and mAP on the COCO dataset will be evaluated.  Next, optimal trade-offs for different task-splitting strategies and compression decisions to optimize total performance will be explored.

Strengths:

1-Important problem. 

2-Good idea

3-Good experiments

4-Good analysis

Weaknesses:

1-Figure 2 had not been used, i.e. referenced at, inside the research

2-Explain how to make sure that the model was not overfitting to the given specifically chosen input data?

3-Lines 394 and 395, at the sentence (tables and figures) make it more clear which tables and figures numbers were meant

4-Line 460, add a separate paragraph that explains the limitations of the proposed method.

5-Line 473, the part (A natural next step…..object detection system.) should be written as a separate paragraph. Add a separate paragraph that explains the limitations of the proposed method.

6-Please can you support us with a place at github, for example, that contains data, results, and other materials used in your research?

Author Response

(The authors gave the same response as above.)

Reviewer 3 Report

It is necessary to remove the inscriptions under the figures. Leave the inscriptions formatted in accordance with the requirements, and remove the inscriptions under the figures. They are simply unreadable. In the work it is necessary to describe in more detail and in detail the method by which the object is determined. The article is similar to material from a textbook for students. The article is too general. This is especially true for figures 1 and 2. They must be removed. Authors submit an article to a scientific publication, rather than write a textbook for students. As a result, due to the lack of scientific results, the provisions in the conclusions are too general and declare banal things. The authors need to take into account the main remark - to bring the material according to the method by which the object is determined. You should also consider issues related to the influence of parameters on the definition of the object. In this form, the article cannot be accepted.

Author Response

(The authors gave the same response as above.)

Reviewer 4 Report

In this paper, a collaborative task method is used to calculate edges in object detection, and a deep learning model is used to optimize execution accuracy and execution time. Experimental results show that using the proposed method can improve system performance and provide real-time response. The research content has some innovation, but the experimental content is incomplete and the theoretical framework is unclear.

1.The summary is not concise enough. The suggestions are as follows: First, point out the problem, then explain the methods used to solve it, and finally use evaluation indicator data to illustrate the effectiveness of the study.

2.In the second section, the overall summary of relevant literature is not in place.

3.In section 3.1 of the literature [29], the author states that "YOLOv5 is a state-of-the-art deep learning object detector", which is incorrect. Please amend it.

4.Legend error. Firstly, "Figure 2" is not quoted in the text; Secondly, in 3.3.1, q [10,95] is specified, while the value of q in Figure 3 is uneven within the range, resulting in no experimental representativeness; Finally, in Figure 3, the annotations should be sorted according to the value of q.

5. The formula is unclear. Lack of explanation of symbols in Equations 1, 2, and 3.

6. The experimental content is not clear. The main content of this article is "Optimizing the transmission delay of images from edge servers to terminal devices". The image compression section should be described in detail, and image display or evaluation indicator data should be added to it.

7.Please explain the specific meaning of "the balance between segmentation and compression decisions" and how to verify this balance.

8. The content structure is missing. In the last paragraph of the first section, the author wrote that he would "summarize the work, acknowledge the limitations, and propose future research directions" in the last section. However, in the last section, the "limitations" were not stated, please add.

In general, it is recommended to reject the manuscript!

Author Response

(The authors gave the same response as above.)

Round 2

Reviewer 3 Report

The reviewers did a good job on the comments. The current version can be accepted for publication.

Reviewer 4 Report

The revision is satisfactory and all concerns from this reviewer has been carefully addressed. It should be publishable now.